# Description of the Exposure of the Most-Followed Spanish Instamoms’ Children to Social Media

**DOI:** 10.3390/ijerph20032426

**Published:** 2023-01-30

**Authors:** Felipe Garrido, Alexandra Alvarez, Juan Luis González-Caballero, Pilar Garcia, Beatriz Couso, Isabel Iriso, Maria Merino, Genny Raffaeli, Patricia Sanmiguel, Cristina Arribas, Alex Vacaroaia, Giacomo Cavallaro

**Affiliations:** 1Department of Pediatrics, Clínica Universidad de Navarra, 28027 Madrid, Spain; 2Department of Statistics and Operations Research, Faculty of Medicine, University of Cadiz, 11003 Cádiz, Spain; 3Jérôme Lejeune Medical Institute, 28012 Madrid, Spain; 4Neonatal Intensive Care Unit, Fondazione IRCCS Ca’ Granda Ospedale Maggiore Policlinico, 20122 Milan, Italy; 5Department of Clinical Sciences and Community Health, Università degli Studi di Milano, 20122 Milan, Italy; 6ISEM Fashion Business School, Universidad de Navarra, 28027 Madrid, Spain

**Keywords:** social media, kids, influencer, Instamoms, sharenting, marketing, Instagram

## Abstract

There is evidence of the risk of overexposure of children on social networks by parents working as influencers. A cross-sectional study of the profiles of the sixteen most-followed Instamoms in Spain was carried out. An analysis of these profiles was performed over a full month (April 2022), three times a week, to describe the representation of influencers’ children in the posts shared by them, as well as their role in the Instamoms’ marketing. A total of 192 evaluations of the profiles were performed in the study period. The average number of children exposed by an Instamom was three, generally preschoolers and schoolchildren. The children appear in a context of the family home and accompanied by their mother. The type of advertising that accompanies the appearance of underage children is usually women or children’s clothing, but also food products, leisure, etc. Appearance of children in the posts had a statistically significant influence on followers measured by the number of likes. Results provided the identification of two Instamom clusters with differentiated behaviors in relation to appearance of children in posts. It is important to involve Social Pediatrics in the protection of the privacy and interests of children given the increase in sharenting. The authors believe that there are concerns about their explicit consent to public exposure from early childhood and about the medium and long-term effect that this may have on their future well-being.

## 1. Introduction

People generating “influence” via social media is an already well-established phenomenon. The presence of “Instamothers” (commonly known as Instamoms) on Instagram and the increase in “sharenting” (sharing + parenting) are topics that provoke an important debate regarding the well-being of their children that warrants the attention of pediatricians. Instamoms are mothers that, despite not being public figures, have public profiles on social media (fundamentally Facebook, Instagram, and YouTube) with thousands of followers, generating influence and posting images and/or videos that include their underage children. Sharenting refers to sharing experiences and, consequently, images and personal information of their children on the Internet [1].

There is evidence of the risk of overexposure of children to social media by parents working as influencers, especially Instamoms, given the number of followers of these profiles on Instagram [2]. From a social and legal point of view, the early exposure of underage children to social media and its monetization, the infringement of their right to privacy, and the lack of legal protection are of concern.

Instagram is a social network launched in 2010 with almost 24,000,000 users in Spain. The most active Spaniards on the platform are between the ages of 18 and39. The network focuses on publishing images and videos that users then “react” to and interact with using written messages. There are also “stories”, which are volatile audiovisual content pieces that can only be visualized during a defined period (generally 24 h), after which they disappear, in contrast to regular posts. It is a social network widely utilized by brands and influencers for commercial/financial purposes [3].

In our research group, we have been concerned about the representation of children in advertising distributed through television or the Internet [4]. In this sense, we have decided that the appearance of children in public Instamom accounts should be a matter to be considered by Social Pediatrics.

Parental sharing of their children’s images or videos raises broader ethical concerns regarding children’s autonomy and right to privacy. Images remain indefinitely publicly online. The circumstances surrounding this phenomenon do not allow minors the capacity to decide on the public dissemination of their image, which leads to a potential unwanted online exposure or overexposure [5]. Definitively, the researchers do not consider that there is a real awareness in society about this problem, and, of course, the regulation by the authorities seems insufficient.

At a general level, there is not enough bibliography about whether there is abuse in the appearance of minors with their parents on social networks. As an example, only recently, and after the design of our study, has a tool that aims to quantify the sharenting phenomenon (Sharenting Evaluation Scale) been validated [6].

Therefore, pediatricians must know whether there is abuse of the image of minors by their parents through social networks, what impact this may have on their later development, and what the potential negative consequences they may generate on their social and personal well-being. Amon et al. highlight the following open questions on this topic: (1) general context in which parental sharing occurs and the parents’ acceptance of the practice, (2) associations between parental sharing and children’s early Internet exposure, and (3) users’ standards for young children’s privacy and autonomy [5]. In this study, we aim to answer, at least in part, these questions, in regard to a specific form of sharenting, the Instamoms.

Our research group has studied in the past the way in which the minors’ images are used in advertisements in print newspapers, on national television channels, and on the Internet. In the Spanish press, in 2008, a child figure appeared in 4.5% of advertisements (NGOs, political promotion, family trips, movies, etc.) [7]. Subsequently, in a non-published analysis of television advertisements, we concluded that the product advertisements that included minors were food, products for children and/or household use, as well as vehicles. Lastly, when we performed a similar analysis of the Internet ads, we concluded that 52% of them where at least one minor appeared were aimed at advertising adult-oriented products (Insurances, NGOs, etc.), and 47.4% contained family-oriented products [4]. 

The objective of this study was to transversally describe the representation of the underage children of the most followed Instamoms in Spain, establish whether they are used when advertising products, and their impact on posts. 

## 2. Materials and Methods

A transversal study was designed in which we followed and analyzed the posts of the most followed Instamoms in Spain. The Instamoms were selected via social media, and the information was available online. The following inclusion criteria were established for the selection of which Instamoms to evaluate: (1) a minimum of 100,000 followers on Instagram (no upper limit was established) and (2) the presence of public Instagram profiles accessible to the researchers. The cut-off points for establishing the minimum number of followers were decided upon by the researchers (PG, FG, AA) through prior monitoring of the most-followed Instamom accounts in Spain. Said cut-off point was random but represented a relevant number for the researchers considering the number of users of the social network in Spain and the general average number of followers of this type of account. Exclusion criteria were not defined. We selected 16 Instamom accounts with such a number of followers.

Any underage individual with an apparent parental relationship to the Instamom that appeared in their publications was considered as an underage family member. For the purpose of the study, all the individuals who were supposedly less than 18 years of age were considered underage. Certainly, from the age of 14, adolescents can consent to appear in images on social networks, but for the researchers, the critical study of sharenting must include these adolescents. We do not have evidence of their consent to the use of their images since they are probably conditioned by their mother’s commercial and advertising activity.

To describe the presence of underage children in the Instamoms’ social networks, we designed the following monitoring methodology. First, an individual pre-selection of the Instamoms to be analyzed was performed by four of the team’s researchers (AA, MM, II, BC). Subsequently, as a group, we agreed upon the final list. The Instamom selection strategy took place through Internet searches, as well as the review of reports from specialized advertising companies in Spain [8]. The four researchers were also responsible for monitoring the 16 selected Instamoms for 4 weeks. All the researchers involved in the follow-up were accustomed to using the social media platform Instagram (©Instagram from Meta), as they are regular users. Therefore, we established two groups of two researchers who had to agree upon the information collected and carry out the evaluation of all available posts jointly on scheduled days. In this study, the term “posts” refers to any uploaded images or videos.

Once the Instamoms were selected, the posts uploaded in each of the 16 accounts were monitored. Follow-up was conducted for 1 month (4 weeks, April 2022). The posts of the Instamoms were analyzed every week, 3 days a week (Tuesday, Thursday, and Sunday). On Tuesday, the posts from Sunday and Monday were reviewed. On Thursday, the posts from Tuesday and Wednesday were reviewed. Finally, the posts from Thursday, Friday, and Saturday were reviewed on Sunday. Variables for the analysis of the posts were established that allowed maximum objectivity (Table 1). As they did not have a validated tool to qualify or quantify sharenting, the researchers decided to compile a very descriptive assessment of the posts, which did not allow the collection of subjective information at all. These variables were defined and agreed upon by the researchers (FG, PG, AA) using the modified Delphi method to enhance the structure of the study. A group of categorical variables was used to describe the presence of children, the mom’s partner’s company, and the advertisement theme in the posts shared by the Instamoms. Due to the volatility of the “stories”, their analysis and relevance for this study was secondary for the researchers, and it was assessed only if they could be viewed (Table 1; V6).

To quantify the impact produced by the posts in the social network of each Instamom, an average number of likes was calculated for each group of days analyzed in three scenarios: (1) when the Instamom appears with children in the posts, (2) when the Instamom appears alone in the posts, (3) when the Instamom appears with another person in the posts that is not a child (adult). For example, if 4 posts with the presence of children had been uploaded between Sunday and Monday, the average of the likes of those 4 posts was calculated on Tuesday. From this average number of likes in each group of days analysed, we have first calculated the average number of likes/day and then the percentage that each Instamom receives in each of the three situations (Table 2).

The collected data were recorded in a spreadsheet (Excel Microsoft) and later imported into the statistical software SPSS v.24 for Windows (IBM SPSS Statistics, Armonk, NY, USA, 2016). A descriptive analysis of the quantitative variables was performed, calculating the mean, median, and range of the values, as well as the frequencies and percentages for the categorical variables. We have analysed the difference in the impact produced by the posts in the three mentioned situations. We have contrasted the normality of the distributions of the percentages and later used the ANOVA contrast and the post hoc contrasts using the DMS method. Finally, the variables used to describe the presence of minors were employed to carry out a cluster analysis with the 16 Instamoms to determine possible differentiated profiles of use of children in the posts, and later the clusters were related to the impact produced by the post.

The ethics committee of the University of Navarra has approved this research study with the code 2022.208. 

## 3. Results

The analysis of the profiles of the 16 Instamoms for 4 weeks produced 192 evaluations. The selected Instamoms had a mean of 544,062 followers (SD ± 565,381) (median 400,000), with a maximum of 2,300,000 and a minimum of 132,000. The mean age of the profiles was 100.3 months (SD ± 24). The mean number of posts published by the moms was 2.5 (SD ± 1.3). All the Instamoms except one (single-parent family) were living as a couple with their children. The mean number of underage children per mother was 3.06 (SD ± 2.17). The mean number of underage children by age group was as follows: infants 0.125 (SD ± 0.34); preschoolers 1.18 (SD ± 0.91); schoolchildren 1.25 (SD ± 1.29); and adolescents 0.5 (SD ± 1.03). In 63 of the 192 profile evaluations, no underage persons were present in any posts (32.8%); in 27 profiles, Instamoms had not published anything in the days analyzed.

We analyzed the environment or context of the posts in which at least one child appeared. In 37 (38.1%) assessments, the photos for the posts were taken in the family home environment, 22 (22.7%) in an outdoor environment, 18 (18.6%) in an urban environment, and 20 (20.6%) in a mixed environment (that is, they appeared in various environmental contexts). Excluding the Instamom who did not appear to live with a regular partner (single-parent family) (n = 181), the mom’s partner rarely appeared in any images or videos uploaded (52/181; 28.7%). Table 1 contains the descriptive parameters of the variables used to describe the presence and situations in which minors appear in Instamom posts.

Table 2 summarizes the impact produced by the posts on the social network of each Instamom. We have included in columns 2 to 4 the average number of likes/day in each of the three evaluated situations: (1) number of likes when at least one child appears, (2) number of likes when the mother appears alone, (3) number of likes when the mother appears in company of an adult. Given the variability of posts and likes received among the 16 Instamoms, we have quantified this impact on each Instamom by the percentage represented by the number of likes/day (with respect to the total number of likes/day) in each of the three situations analysed (columns 6 to 8). To compare the impact produced by the posts between the three evaluated situations, we have compared the average percentages of likes/day of each Instamom. We have first contrasted normality with the Shapiro–Wilk test, obtaining values of *p* = 0.106, *p* = 0.156, *p* = 0.456, respectively, in each of the three evaluated situations, which indicates that we can use the contrast parametric ANOVA to compare the distributions of likes. With this ANOVA contrast, we obtained a value of *p* = 0.017, which means that there are significant differences in the impact received depending on the three situations. The post hoc contrasts using DMS have obtained a significant difference (*p* = 0.048) between the situation (1) and (2), with IC95% (1–2) = (0.15–28.57); and a significant difference (*p* = 0.006) of (1) vs. (3), with 95%CI (1–3) = (6.28–34.71).

We have also analysed the profiles of the Instamoms regarding the variables related to the presence of situations in which a child appears (V1, V3, V5 and V6, from Table 1). Table 3 contains these variables, and the last column containsthe belonging of each Instamom to each one of the two clusters obtained with the hierarchical method using the squared Euclidean distance.

Figure 1 graphically represents the profiles of the Instamoms. It can be seen that the frequency of use of minors is lower in the profiles of the Instamoms of cluster 1 (blue lines) compared to the profiles of the Instamoms of cluster 2 (yellow lines). Finally, we compared the impact of the posts between the two clusters, obtaining significance (*p* = 0.030) in the case of the situation in which at least one minor is present, but not in the other two situations in which the mother appears alone (*p* = 0.091) or the mother appears in the company of other adults (*p* = 0.118).

## 4. Discussion

Social media is the present and the future of communication. In Spain, more than 134,000 influencers with more than 1000 followers on average post content on social networks such as YouTube, Facebook, Instagram, or Twitch. Of these, more than 7500 live off the income generated by creating digital content and its dissemination on social networks. Advertisement of products by influencers is considered one of the most effective marketing and sales strategies for the Millennial, Z, and Alpha generations [9]. 

Sharenting is the parents’ public exposure of the children on social media. When this exposure is exaggerated, we refer to it as “oversharenting”. This implies the exposure of personal information of underage children, especially photographs and videos, on social media [10]. This can sometimes mean financial compensation, which can become the family’s main income source [11]. Moreover, sharenting creates a premature digital identity, even before birth (digital birth) [12]. This raises questions as the information parents share leaves digital footprints, and the parents (unconsciously) contribute to their child’s construction of online identity [13]. Adolescents feel specifically concerned about sharenting. Ouvrein et al. in a qualitative study share the opinion that (1) not all kinds of photos can be shared; (2) the shared information cannot be too personal; (3) parents should restrict their sharing behavior in certain occasions (such as everyday activities) [14]. Sharenting thus constitutes a challenge for legislators as well as for social pediatrics.

The General Data Protection Regulation 2016/679 of the European Parliament and the Council of 27 April 2016 allows each EU member country to lower the age of consent, which is 16, to a minimum of 13 years. If the child is under this age, such treatment is only considered lawful if the consent was authorized by the holder of parental authority [15]. In Spain, Organic Law 3/2018 of December 5 establishes that the processing of personal data may depend on a minor when they are at least 14 years old. The Law adds that “the controller will make reasonable efforts to verify that consent to the processing of data in children under 14 years of age was given or authorized by the holder of parental authority, taking into account the available technology” [16]. On the other hand, Instagram consents to a minor’s registration if they are at least 13 years old [2]. In any case, underage children have the fundamental rights to personal and familial privacy and control over their own image, which is inherent to every person, regardless of their age or ability to act.

Instagram is the social network where the use of photographs by Instamoms with their children has increased the most due to the large number of interactions generated by this content and the promotion of products for children [2]. The Instamoms usually begin sharenting from pregnancy to raising children [12]. Some do it in addition to the central theme of their Instagram account, and others focus their account on sharing this content about the pregnancy and growth of their children. Instamoms have gained relevance in the research fields of communications and commercial research, given their legal and ethical implications, although most of these studies have focused on the study of advertising transparency [17]. The work of Instamoms is a balance between financial gain and community building where these influencers aim to provide their followers with enough content to maintain a supportive digital community of mothers, show authenticity, and maximize the platform as a source for monetary revenue [18]. In a recent published thesis, Kojok et al. analyze the participation of children in a contemporary home digital labor economy [19]. They state that the interaction between entrepreneurship and social media has resulted in a specific strand of child labor that should be scrutinized.

Posting images of children has consequences for them. Hiniker et al. state that in the United States, minors find content uploaded by their parents embarrassing and feel frustrated with the posts publicly contributing to facilitation of their presence on social networks without their permission [20]. Kopecky et al. affirm that sharenting can also have the negative aspect of using underage children as a commercial tool [21]. In Spain, 19% of underage children between the ages of 9 and 17 state that their parents have published their images without permission, according to a recent study by Garmendia et al. [22]. In this study, this percentage is increased for adolescents (14% vs. 23%). The discomfort caused by sharenting is also greater in adolescents than in non-adolescents, and even more so if they are female (13% vs. 4%).

There are no specific tools that allow quantifying the degree of sharenting of an Instagram user. As such, our research may show this methodological limitation. However, our analysis has been purely descriptive and established simple and objective variables attempting to avoid any subjective effect of evaluator. Recently, after designing and carrying out our study, Romero-Rodríguez et al. have developed and validated a scale (Sharenting Evaluation Scale) that can fulfill said evaluative function [6]. In any case, as authors, we think that the development of validated evaluation tools for Instagram accounts to determine their potential effects on minors as well as the abuse of their image should be considered as research objectives in Social Pediatrics. 

A recent study analyzing the Instagram activity of 10 Spanish influencers established that 45.6% of the posts appeared included an underage child, and these posts had 41% more likes than those where a child does not appear [23]. In our study, the percentage of publications in which at least one child appears is 58.7%, with the appearance of the mom’s partner being even lower (Table 1). This difference may be due to methodological differences. In our study, we recorded and prospectively analyzed the activity of the Instamom accounts three times a week. Jiménez-Iglesias et al. do not describe the way in which they analyze the activity in their study, so comparing results can be inadequate. Still, they seem to have carried out a retrospective analysis of each of the publications, taken globally.

The analysis represented in Table 2 shows that the number of interactions with the posts of the Instamoms increases with the appearance of the minor. As will be discussed later, this is an important conclusion from the point of view of Social Pediatrics. Our confirmed hypothesis about the summative effect of the presence of sons and daughters in the images uploaded by Instamoms should not be trivialized. The presence of children supposes a greater visualization of posts and therefore could lead to a risk of overuse of their image.

In our study, we managed to establish two representative clusters of the Instamoms. We were able to define these groups based on the presence of the minor in the posts. The identification of two Instamom clusters is a relevant conclusion for the researchers since it allows us to determine that there are two potential and differentiated behaviors, one in which there is a greater overexposure of the children, and another which is more conservative and theoretically protective of the children’s image. In either of the two cases, the simple exposure of minors on the social network is debatable, but this statistical standardization makes it possible to establish that there are different ways of using said image. In the cluster of accounts that used the minor more regularly, we were also able to verify that their appearance had a greater impact in the form of likes on their posts. This denotes the promotional effect that minors have on the accounts of the Instamoms, which surely conditions a sometimes-abusive exposure of the children. In summary, this supposes a reinforcing of the effect of the minor in the images uploaded by their mothers. This conclusion seems relevant to the authors and should continue to be studied by researchers, not only on Instagram, but on any social network for present or future use.

Jorge et al., in a qualitative study of Portuguese Instamoms, described that the profiles that were created between 2011 and 2015, coinciding with a period of economic difficulties, after a period of time began to monetize their activity as influencers [1]. In our study, the Instamoms analyzed mainly advertise both women’s and children’s clothing. To a lesser extent, they advertise food products, leisure, etc. Unfortunately, we have yet to obtain a bibliography to compare these results in our environment. However, in a recent study, Monge-Benito et al. analyzed the marketing strategies of influencers and established maternity, fashion, cosmetics, and fitness as the main advertising areas [24]. In addition, they concluded that the Instamoms are the subset of influencers that most frequently label their advertising activity as a paid promotion using a hashtag, which can be considered an act of transparency. However, we have yet to analyze this practice.

Our study has some limitations. Firstly, as previously mentioned, the absence of a validated tool to quantify sharenting can lead to a certain subjectivity in the methodology, which has been offset by the type of variables chosen. Secondly, the evaluation of the Instamom accounts was performed in a single month of the year, and not in several, which could have motivated a variability in the commercialization of products (Black Friday, Christmas, etc.) and potentially a greater participation of children. Thirdly, since it is a purely descriptive study, it does not evaluate the impact of this exposure on minors’ social networks, in the short or long term, although, in any case, it was not an objective of this study.

## 5. Conclusions

In conclusion, European experts condemn the lack of a complete and clear legal framework that regulates the appearance of underage children in Instagram posts shared on social media, specifically on Instagram accounts run by their parents. The way to protect the privacy and interests of children remains to be determined, given the increase in sharenting. In our study, we described that the most-followed Instamoms in Spain share images of an average of three of their children, normally in the context of the family home and accompanied by their mother. The appearance of the mom’s partner is infrequent. When at least one child appears in the posts, the mother advertises products (generally clothing) in slightly less than half of them. Appearance of children in the posts has a statistically significant influence on followers measured by the number of likes. From our point of view, more restrictive regulations must be established on the use of children by influencers, and the possible short- and long-term effects of overexposure should be determined.

## Figures and Tables

**Figure 1 ijerph-20-02426-f001:**
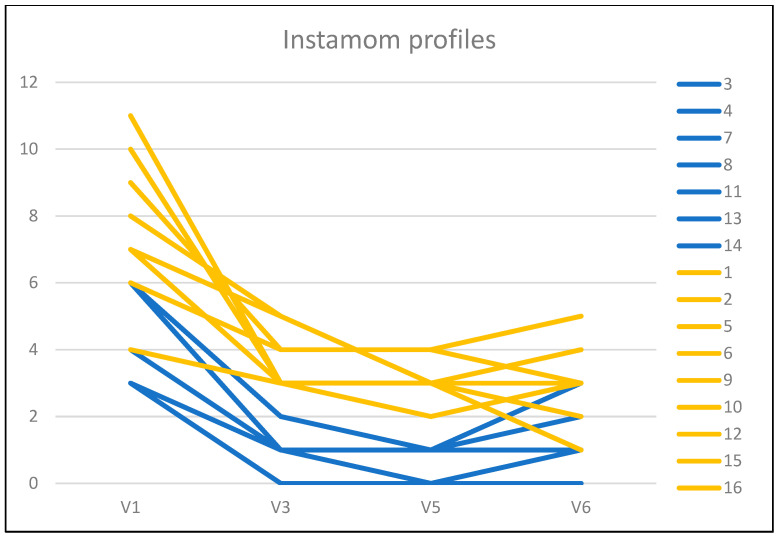
Graphical representation of Instamom cluster profiles.

**Table 1 ijerph-20-02426-t001:** Summary of the variables used to describe the presence and situations in which children appear in the Instamom posts.

Variables	Results n (%)
V1: What is the environment of the posts in which one or more children appear?	Home 37 (38.1) Urban 18 (18.6) Nature 22 (22.7) Mixed 20 (20.6)
V2: Is the mom´s partner present in the posts?	No 129 (71.3) Yes 52 (28.7)
V3: Does any child appear in a post with advertisement?	No 150 (78.1) Yes 42 (21.9)
V4: Who is with the child when appearing in posts?	Alone 4 (4.1) With mother 47 (48.5) With family/others 46 (47.4)
V5: Is the child actively using the product that is being advertised?	No 159 (82.8) Yes 33 (17.2)
V6: In the cases where a story was seen, was the child used to advertise a product?	No 158 (82.3) Yes 34 (17.7)

**Table 2 ijerph-20-02426-t002:** Average number of likes/day and percentages they represent for each Instamom in each of the three analysed situations: (1) when at least one child appears, (2) when the mother appears alone, (3) when the mother appears in company of an adult.

Instamoms	Likes/Day (1)	Likes/Day (2)	Likes/Day (3)	Total Likes/Day	% Like/Day (1)	% Like/Day (2)	% Like/Day (3)
I1	20,613.71	26,621.71	31,622.46	78,857.89	26.14	33.76	40.10
I2	23,156.07	3074.46	1338.32	27,568.86	83.99	11.15	4.85
I3	1043.36	950.86	2951.25	4945.46	21.10	19.23	59.68
I4	6068.14	2935.36	372.43	9375.93	64.72	31.31	3.97
I5	1998.00	836,96	338.50	3173.46	62.96	26.37	10.67
I6	4253.71	1308.93	1566.64	7129.29	59.67	18.36	21.97
I7	3164.25	6142.25	2925.36	12,231.86	25.87	50.22	23.92
I8	2229.46	1768.93	2388.75	6387.14	34.91	27.70	37.40
I9	2064.00	3617.14	2834.14	8515.29	24.24	42.48	33.28
I10	3500.36	569.75	323.29	4393.39	79.67	12.97	7.36
I11	550.07	3092.57	728.25	4370.89	12.58	70.75	16.66
I12	1829.14	315.57	17.71	2162.43	84.59	14.59	0.82
I13	1104.57	852.86	920.36	2877.79	38.38	29.64	31.98
I14	749.25	2551.61	3699.64	7000.50	10.70	36.45	52.85
I15	1701.00	2091.86	1822.71	5615.57	30.29	37.25	32.46
I16	2887.71	1323.75	648.75	4860.21	59.42	27.24	13.35
Mean ± sd	4807.05 ± 6829.15	3628.41 ± 6307.41	3406.16 ± 7610.37	11,841.62 ± 18,844.52	44.95 ± 25.48	30.59 ± 15.25	24.46 ± 17.67
Median (range)	2146.73 (550.07–23,156.07)	1930.40 (315.57–26,621.71)	1452.48 (17.71–31,622.46)	6001.36 (2161.43–78,857.89)	36.65 (10.70–84.59)	28.67 (11.15–70.75)	22.95 (0.82–59.68)

**Table 3 ijerph-20-02426-t003:** The assignment of the Instamoms to two clusters depending on the quantification of the variables represented in Table 1.

Instamom Code	(V1)	(V3)	(V5)	(V6)	Clusters
I3	3	0	0	0	1
I4	6	1	1	3	1
I7	3	1	1	1	1
I8	6	2	1	3	1
I11	3	1	1	2	1
I13	4	1	0	1	1
I14	3	1	1	1	1
I1	4	3	3	3	2
I2	10	3	3	4	2
I5	8	5	3	1	2
I6	9	4	4	5	2
I9	6	4	4	3	2
I10	7	5	3	1	2
I12	11	3	2	3	2
I15	7	5	3	2	2
I16	7	3	3	1	2

## Data Availability

Not applicable.

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
