# Peer review of "Description of the Exposure of the Most-Followed Spanish Instamoms’ Children to Social Media"

_ijerph, 2023, doi:10.3390/ijerph20032426_

Round 1
Reviewer 1 Report (Previous Reviewer 1)
The authors have improved their first version of the paper and incorporated all mentioned comments.
Author Response
thank you so much
Reviewer 2 Report (Previous Reviewer 2)
The author revised the article successfully. I have no additional comment. I wish authors continued success in their future endeavours.
Author Response
Thank you very much
Reviewer 3 Report (New Reviewer)
Dear Authors, I congratulate you on the research idea - it is very interesting. However, I have a few comments and suggestions, which I have included in the file below. I hope that my comments will be useful to improve the article.

Author Response
Thanks for your suggestions. Please see cover letter

This manuscript is a resubmission of an earlier submission. The following is a list of the peer review reports and author responses from that submission.
Round 1
Reviewer 1 Report
There is a substantial lack of literature review in this paper.
There should be an Ethics Committee approval.
The analysis is more descriptive and would need better statistical analysis. Kruskal-Wallis analysis does not bring any added value.
The discussion is very short, and not many surprising outputs can be found from it.
There is no part on the limitations of the study.
Reviewer 2 Report
Many thanks for providing me with the opportunity to read your interesting study. I have some suggestions in order to help you improve your article, and wish you the best of luck in this regard.
- First, I would like to see the gaps in the introduction section emphasized. What prompted you to perform this research? What are the unanswered questions in the existing literature, and how will you address to these queries? Please emphasize the originality and uniqueness of the study. I do not think you answered the three important questions that an effective research motive/objective should answer (Grant & Pollock, 2011): Who cares? What do we know, what don't we know, and so what? And what will we learn? Please pay attention aforesaid points in the introduction section.
- As the authors stated, transversal study was designed in which they followed and analyzed the posts of 60 the 16 most followed Instamoms in Spain.
To ensure that this study was conducted in a scientific manner, the authors need to prove that both data collection and interpretation of the analysis were made scientifically. Regarding the data collection, the authors may be able to clarify their search strategy, as researchers typically do in a high quality study.
- There are a few too many unsubstantiated/uncited statements - please be careful with this. Theoretical implications are light. I would expect to see greater attention paid to demonstrating the theoretical contribution of your work. Please revisit this.